∂ | **Open Peer Review** | Systems Biology | Research Article

# Proteome efficiency of metabolic pathways in *Escherichia coli* increases along the nutrient flow

Xiao-Pan Hu,[1,2] Stefan Schroeder,[1,2] Martin J. Lercher[1,2]

**ABSTRACT**   Understanding the allocation of the cellular proteome to different cellular processes is central to unraveling the organizing principles of bacterial physiology. Proteome allocation to protein translation itself is maximally efficient, i.e., it represents the minimal allocation of dry mass able to sustain the observed protein production rate. In contrast, recent studies on bacteria have demonstrated that the concentrations of many proteins exceed the minimal level required to support the observed growth rate, indicating some heterogeneity across pathways in their proteome efficiency. Here, we systematically analyze the proteome efficiency of metabolic pathways, which together account for more than half of the *Escherichia coli* proteome during exponential growth. Comparing the predicted minimal and the observed proteome allocation to different metabolic pathways across growth conditions, we find that the protein abundance in the most costly biosynthesis pathways—those for amino acid biosynthesis and cofactor biosynthesis—is regulated for near-optimal efficiency. Overall, proteome efficiency increases along the carbon flow through the metabolic network; proteins involved in pathways of nutrient uptake and central metabolism tend to be highly over-abundant, while proteins involved in anabolic pathways and in protein translation are much closer to the expected minimal abundance across conditions. Our work thus provides a bird's-eye view of metabolic pathway efficiency, demonstrating systematic deviations from optimal cellular efficiency at the network level.

**IMPORTANCE**   Protein translation is the most expensive cellular process in fast-growing bacteria, and efficient proteome usage should thus be under strong natural selection. However, recent studies show that a considerable part of the proteome is unneeded for instantaneous cell growth in *Escherichia coli*. We still lack a systematic understanding of how this excess proteome is distributed across different pathways as a function of the growth conditions. We estimated the minimal required proteome across growth conditions in *E. coli* and compared the predictions with experimental data. We found that the proteome allocated to the most expensive internal pathways, including translation and the synthesis of amino acids and cofactors, is near the minimally required levels. In contrast, transporters and central carbon metabolism show much higher proteome levels than the predicted minimal abundance. Our analyses show that the proteome fraction unneeded for instantaneous cell growth decreases along the nutrient flow in *E. coli*.

**KEYWORDS**   resource allocation, proteome efficiency, growth rate, growth law, metabolic pathways, biosynthetic pathways, central carbon metabolism, glyoxylate shunt

Proteins account for more than half of the cell dry mass in *Escherichia coli* (1) and drive most biological processes. How and why proteome is allocated to different cellular processes and pathways is a vital question for understanding the principles

Address correspondence to Xiao-Pan Hu, xiao-pan.hu@hhu.de, or Martin J. Lercher, Martin.lercher@hhu.de.

The authors declare no conflict of interest.

See the funding table on p. 17.

behind bacterial physiology (2). Proteome allocation into different groups of genes is growth rate-dependent (3). When partitioning the proteome into specific, coarse-grained "sectors", the corresponding proteome fractions follow simple, empirical growth laws, increasing or decreasing linearly with the growth rate $\mu$ (4–7). For example, the proteome fraction allocated to the ribosome and ribosome-affiliated proteins [the R-sector (6)] scales as a linear function of growth rate under nutrient-limiting conditions (4).

Why does the proteome composition scale with the growth rate? Protein is the most abundant and costly macromolecule in bacterial cells. It has thus been speculated that the proteome composition is adjusted to the specific growth condition to maximize the growth rate (8). If this were true, all protein concentrations would be at the minimal level required to sustain the observed cellular growth rate. This simple assumption has been widely used in computational models of cellular growth (9–15). However, even if proteome allocation had evolved to be maximally efficient, it is not obvious that this efficiency would simply maximize the instantaneous growth rate. Instead, it appears likely that proteome allocation has evolved to maximize cellular fitness in unpredictable, dynamic environments with varying nutrients and involving periods of famine and stresses (8). Indeed, recent experimental work indicates that the proteome is not allocated to achieve maximal efficiency in unevolved *E. coli* strains, at least not in the naïve sense of maximizing the instantaneous growth rate. First, a large fraction of the observed proteome is unneeded for the current environment, especially at low growth rates (16). Second, the total *E. coli* protein concentration remained approximately constant in chemostat growth on a minimal medium with glucose, despite growth range varying between 0.12 h$^{-1}$ and 0.5 h$^{-1}$ (17). Third, the fluxes through some cellular processes, e.g., nutrient transport and energy production, are not limited by specific proteins in these pathways at low growth rates (18). Fourth, growth rate can increase by approximately 20% over a few hundred generations in adaptive laboratory evolution experiments on minimal media (19), a process associated with reductions in the abundance of unused proteins (16). Thus, there is ample evidence that *E. coli* proteome allocation is not globally optimized for maximizing the instantaneous growth rate.

On the pathway level, however, proteome allocation to at least one cellular process—protein translation—is optimized for maximal efficiency at the given protein synthesis rate (18, 20–22). This indicates that while the global allocation of proteins is not always optimized for maximal growth rate, the proteome allocation to some cellular pathways is at a local optimum, i.e., the individual pathway utilizes the minimal protein mass required to support the observed pathway output. In contrast, proteome allocation to transporters scales contrary to the optimal demand with decreasing growth rate in *E. coli*; at increasingly lower growth rates, bacterial cells harbor more and more transporters for nutrients that are currently not available (17, 18).

Why do cells optimize resource allocation to certain pathways (translation) but not others (transporters)? From a cellular pathway topology perspective, transporters are located at the interface with the environment, while proteins for translation are located at the end of nutrient flow. Bacteria such as *E. coli* are living in constantly changing environments, but have only a very limited ability to sense external nutrient levels. Therefore, transporters should not only transport enough nutrients for cell growth under the current conditions but also allow the cell to quickly import alternative substrates that become available in upcoming conditions. To maximize fitness across changing environments, it is plausible that bacteria growing on preferred nutrients should invest much of their resources into proteins required for instantaneous growth, while bacteria growing on unpreferred nutrients should allocate more resources to the preparation for future environments. Unlike transporters, translation proteins are located in the interior of cellular processes and rarely have direct connections to the environments. Moreover, in contrast to sensing the large number of potential nutrients and their combinations, sensing an increased or decreased demand for protein production is essentially a one-dimensional problem. Thus, the cell might have evolved a simple and efficient

way to regulate the proteome allocation to the translation machinery to the minimal required level for instantaneous cell growth, rather than making them dependent on specific nutrients. Indeed, in *E. coli*, the ribosomal genes are mainly regulated by the concentration of a single molecule species, ppGpp (23, 24).

Based on these observations, we speculated that more generally, the proteomic efficiency of pathways might depend on their positions in the metabolic network. We hypothesized that proteome efficiency—defined as the ratio between minimally required and observed protein concentrations—increases along the carbon flow, from transporters at the network periphery to translation at the network core. In *E. coli* growing on minimal media with different carbon sources, more than half of the proteome by mass are metabolic enzymes (17). Computational models can predict the optimally efficient proteome allocation to each metabolic pathway (9, 12, 14, 16), and quantitative proteomics data are available for *E. coli* growing on a wide range of minimal media with different carbon sources (17). To test our hypothesis, we exploit these resources to compare experimental data across diverse minimal carbon media conditions (17) to the predicted optimal proteome allocation to pathways at the observed growth rate. As expected, we find that pathways differ systematically in how much excess protein mass is allocated to them compared to the local optimum, with decreasing excesses over optimal allocation along the carbon flow from nutrient import to protein production.

## RESULTS AND DISCUSSION

### Modeling proteome allocation with linear enzyme kinetics and growth rate-dependent biomass composition

To investigate the local pathway efficiency, we predicted the local optima of all metabolic enzymes in the *E. coli* genome-scale model *i*ML1515 (25) using MOMENT (MetabOlic Modeling with ENzyme kineTics) (9, 26, 27). MOMENT is a version of flux balance analysis (FBA) that incorporates approximate enzyme kinetics, using effective turnover numbers to estimate the enzyme amount required to support a given reaction flux. In its standard application, MOMENT attempts to find the maximal rate of biomass production given a constraint on the total proteome fraction allocated to enzymes and transporters (9, 15).

While the *i*ML1515 model only provides a single biomass reaction with fixed stoichiometry, the RNA/protein mass ratio (4) and the cell surface/volume ratio (28) of *E. coli* have been observed to change across growth rates. To facilitate accurate predictions of cellular resource allocation across growth rates, we adjusted the biomass reaction to reflect the observed growth rate dependence of the production of RNA, protein, and cell envelope components [murein, lipopolysaccharides (LPSs), and lipid; see Materials and Methods, Fig. S1; Table S1]. To estimate the influence of this growth rate dependence, we compared our results with those obtained from the original, growth rate-independent biomass reaction.

MOMENT estimates the enzyme concentration required to support a given flux $v_i$ as $[E_i] = v_i/k_i$, where the effective turnover number $k_i$ of the enzyme $E_i$ is assumed to be constant across conditions. While this linear formulation ignores changes in the saturation of the enzyme due to changing metabolite concentrations, it still provides a useful approximation to the true growth rate dependence (29). An important factor in this type of model is the choice of effective turnover numbers $k_i$. In this work, we parameterized reactions of the *i*ML1515 model with three types of $k_i$. Where available, we used experimental measurements of maximal *in vivo* effective enzyme turnover number ($k_{app,max}$); these have been shown to represent turnover in the cellular environment more accurately than *in vitro* estimates of enzyme turnover numbers ($k_{cat}$) (30, 31). We obtained the values of $k_{app,max}$ from Heckmann et al. (31), who estimated these values by using proteomics and fluxomics data across 21 evolved *E. coli* strains (31). The corresponding $k_{app,max}$ estimates are highly consistent with those of an independent study on wild-type *E. coli* across multiple growth conditions (30, 31). For those enzymes for which *in vivo* $k_{app,max}$ estimates were unavailable, we used *in vitro* $k_{cat}$ estimates

collected by Adadi et al. from public databases (9), if these were available. If neither *in vivo* $k_{app,max}$ nor *in vitro* $k_{cat}$ estimates were available, we used maximal *in vivo* enzyme turnover numbers predicted using machine learning ($k_{app,ml}$), which were estimated by Heckmann et al. using enzyme structures, enzyme network context, and biochemical mechanisms as input features (31). By proteome mass, approximately 40% of reactions were parameterized with $k_{app,max}$, 39% with $k_{cat}$, and 15% with $k_{app,ml}$ (Fig. S2; Table S2).

Based on these enzyme turnover numbers and the growth rate-dependent biomass function, we used MOMENT to identify the minimal total mass concentration of enzymes and transporters that can support the observed growth rate on a given carbon source (in units of gram per gram of dry weight, $g/g_{DW}$; the predicted and measured concentrations of individual proteins are listed in Table S3; for more details, see Materials and Methods). Thus, our predictions do not reflect the resource allocation that would lead to the highest growth rate in a given environment (global optimality), but instead quantify the minimal proteome allocation into pathways required to sustain the observed growth rate (local optimality). Note that all effective turnover numbers used for model parameterization aim to approximate the maximal enzyme turnover numbers $k_{cat}$; hence, the model estimates of enzyme concentrations are those that would be required to support a given flux if all enzymes were fully saturated with their substrates. Accordingly, our model estimates provide a lower bound for proteome allocation into pathways, which is expected to underestimate the actual demand, especially at lower growth rates (29). For comparison, we also report the results of calculations assuming expected growth rate-dependent enzyme saturation levels in the following sections.

## Proteome efficiency increases along nutrient flow in coarse-grained pathways

Following earlier work (16), we first compared the predicted minimal required proteome with experimental data across the complete metabolic network, focusing on minimal media with different carbon sources. As *E. coli* uses different central metabolic reactions for growth on glycolytic and gluconeogenic carbon sources, and as most of the proteome data in reference (17) were measured on glycolytic carbon sources, we focus on the proteome efficiency of metabolic pathways on glycolytic carbon sources here; results for gluconeogenic carbon sources are shown in Table S4. We classified proteins into three groups on the basis of their experimental and predicted proteome allocation. An individual protein is labeled as follows:

- "shared" if its presence is predicted under local optimality and is confirmed in the experiment [these proteins were labeled "utilized" in reference (16)];
- "measured-only" if it is found in the experiment but predicted to be absent [these proteins were labeled "un-utilized" in reference (16)];
- "predicted-only" if its presence is predicted but not confirmed in the experiment.

The predicted-only proteins account for only a very small fraction of the total predicted proteins (<1%) in all studied pathways, except for nutrient transport and proteins without assigned pathways in this study ("others") (Fig. S3). We thus do not include the predicted-only proteins in the following figures.

Metabolic enzymes account for a decreasing fraction of the proteome with increasing growth rate, with observed proteome fractions ranging from 67% to 53% (Fig. S4). In agreement with earlier work (16), we found that the total abundance of shared proteins—those required for maximally efficient growth—increases with growth rate, but far exceeds the predicted globally optimal abundance, especially at lower growth rates (Fig. S4).

To assess the pathway-specific proteome efficiency, we examined the following four aspects.

1. For a given pathway, we summed the mass concentrations of all shared proteins—those that are predicted to be active and found experimentally—in each growth condition for both the observed proteins and for the locally optimal prediction. We then calculated the Pearson correlation coefficient $r$ between the two combined mass concentrations across conditions (denoted as $r_{pathway}$). For locally optimal proteome allocation and if the assumption of constant enzyme saturation would hold, this correlation should approach $r = 1$. Importantly, this expectation holds independent of the values chosen for the enzyme kinetic parameter values.

2. The geometric mean fold error (GMFE) of predicted vs observed protein concentrations of the pathway's shared proteins (denoted as $GMFE_{pathway}$), calculated across proteins and growth conditions. The GMFE shows by which factor the observed concentrations deviate from predicted values on average.

3. The experimentally observed mass fraction of measured-only proteins of the pathway in a given growth condition (denoted as $f_{measured-only}$). This is the proteome fraction that makes no contribution to growth according to our predictions.

4. The squared Pearson's correlation coefficient between predicted and measured abundances across individual proteins in a given growth condition (denoted as $r_{individual}^2$). While measures (1)–(3) assess optimality at the pathway level, this last measure quantifies the relationships between proteins within the pathway: a correlation coefficient close to 1 indicates that all proteins are equally close to—or equally distant from—the optimal prediction. Note that in contrast to measure (1), the comparison across individual proteins relies strongly on the accuracy of the individual turnover numbers. As the latter are only known approximately, we expect these estimates to be noisy.

Table 1 shows the pathway proteome efficiency measures on glycolytic carbon sources, which are discussed in the following subsections.

To test if the proteome efficiency of pathways increases with carbon flow, we first partitioned the metabolic proteins in the iML1515 model into four coarse-grained sets (see Materials and Methods, and Tables S5 and S6 for pathway membership): (1) transporters, which shuttle metabolites across the outer or inner membrane; (2) central metabolism, which produces precursor metabolites and energy for all other cellular processes; (3) biosynthesis pathways, which utilize precursors and energy generated by central metabolism to produce building blocks of macromolecules; and (4) other enzymes, that is, all enzymes in the iML1515 model not included in (1)–(3) (denoted as "others"; these proteins are not assigned to a specific position along the nutrient flow). The iML1515 model does not include a representation of translation processes. To provide a more complete bird's-eye view of nutrient flow, we also included, in our analyses, the proteome efficiency of the translation machinery (predicted and measured proteome allocation to the ribosome, elongation factor Tu, and elongation factor Ts). In contrast to that of the other pathways, the proteome efficiency for translation was not calculated with the MOMENT model described above, but was directly obtained from our previous work (20), which utilized the same proteomics data analyzed here (17). It should be noted that the input of the translation model did not enforce any growth rate-dependent biomass composition (in particular, a specific RNA/protein ratio); the model predicted the resources allocated to translation by minimizing the total mass concentration of all translational components at the required protein synthesis rate, thereby dynamically adjusting the relative allocation of biomass to RNA and protein (20).

In these coarse-grained pathways, carbon and other nutrients flow from transporters to central metabolism to biosynthesis pathways to translation. For all four aspects assessed, the proteome efficiency gradually increases along the nutrient flow (Fig. 1): $r_{pathway}$ increases from −0.75 to 0.93 (Table 1 lists the fraction of variance explained by this variable, $r_{pathway}^2$); $GMFE_{pathway}$ decreases from 3.39 to 1.35; $f_{measured-only}$ decreases from 0.92 to 0; and $r_{individual}^2$ increases from 0.13 to 0.98 (Table 1).

**TABLE 1** Proteome efficiency of pathways

| Pathway | Pathway expression (for shared proteins) (n = 14)[a] | | | Measured-only fraction[b] (median across 14 conditions) ($f_{measured-only}$) | Individual shared proteins; median across 14 conditions[c] | | |
|---|---|---|---|---|---|---|---|
| | $r_{pathway}^2$ | P | $GMFE_{pathway}$ | | $r_{individual}^2$ | P | $n^d$ |
| Measures (1)–(4) | (1) | (1) | (2) | (3) | (4) | (4) | |
| **Transporters[f]** | **(−) 0.57[e]** | **$1.9 \times 10^{-3}$** | **3.39** | **0.92** | **0.13** | **0.64** | **4** |
| **Central metabolism** | **0.024** | **0.60** | **2.32** | **0.31** | **0.15** | **$3.3 \times 10^{-3}$** | **56** |
| Glycolysis | 0.63 | $6.9 \times 10^{-4}$ | 2.21 | 0.08 | 0.35 | 0.05 | 11 |
| Pentose phosphate pathway | 0.72 | $1.3 \times 10^{-4}$ | 1.30 | 0.39 | 0.32 | 0.24 | 6 |
| Tricarboxylic acid (TCA) cycle | (−) 0.43[e] | 0.01 | 6.40 | 0.10 | 0.38 | 0.03 | 12 |
| Glyoxylate shunt | _[g] | _[g] | _[g] | 1 | _[g] | _[g] | 0 |
| Energy generation | (−) 0.02[e] | 0.61 | 1.63 | 0.06 | 0.11 | 0.08 | 28 |
| Central metabolism, others | 0.44 | $9.4 \times 10^{-3}$ | 1.56 | 0.55 | 0.98 | 0.10 | 3 |
| **Biosynthesis** | **0.84** | **$4.8 \times 10^{-6}$** | **1.70** | **0.26** | **0.45** | **$4.2 \times 10^{-31}$** | **226** |
| Amino acid | 0.77 | $3.7 \times 10^{-5}$ | 1.40 | 0.30 | 0.45 | $1.1 \times 10^{-10}$ | 72 |
| Nucleotide | 0.67 | $3.7 \times 10^{-4}$ | 3.32 | 0.23 | 0.15 | 0.05 | 28 |
| Cell envelope | 0.43 | 0.01 | 1.88 | 0.14 | 0.38 | $2.3 \times 10^{-5}$ | 40 |
| Cofactor | 0.84 | $4.9 \times 10^{-6}$ | 1.24 | 0.11 | 0.59 | $4.1 \times 10^{-15}$ | 72 |
| Biosynthesis, others | 0.60 | $1.1 \times 10^{-3}$ | 2.91 | 0.25 | 0.46 | $5.4 \times 10^{-5}$ | 29 |
| **Translation** | **0.87** | **$1.1 \times 10^{-6}$** | **1.35** | **0** | **0.98** | **0.08** | **3** |
| **Others (other metabolic enzymes)** | **0.004** | **0.84** | **1.79** | **0.91** | **0.16** | **0.03** | **30** |
| **Total metabolism** | **0.72** | **$1.4 \times 10^{-4}$** | **1.79** | **0.52** | **0.35** | **$1.7 \times 10^{-30}$** | **309** |

[a]Values reflect the local optimality of complete pathways across conditions. n = 14 indicates the number of glycolytic carbon sources analyzed.
[b]Mass fraction of measured-only (unpredicted but observed) proteins relative to all proteins in the pathway.
[c]These columns reflect the local optimality compared across individual proteins within each pathway at a given growth condition; values are medians across the n = 14 glycolytic growth conditions.
[d]Number of proteins in each pathway or pathway set.
[e]Negative correlation coefficient $r_{pathway}$.
[f]Bold font indicates coarse-grained pathways.
[g]–, not applicable.

As explained above, our estimates provide a lower bound for the required protein abundances, as they are calculated under the assumption that all enzymes are fully saturated with their substrates. Previous work argued that the optimal metabolite concentration and hence the optimal enzyme saturation level increase with increasing growth rate (29). We thus estimated the expected enzyme saturation levels in our studied conditions (see Materials and Methods) and used them to re-estimate the minimal required proteome. Despite changes in the numerical values, all trends in the new results are consistent with those based on the original estimates (see the comparison in Fig. S5). Note that about 55% of enzymes by mass are parameterized by maximal *in vivo* effective turnover numbers in our model, which already reflect non-perfect saturation levels (30, 31). Thus, the true optimal concentration of enzymes is likely located between the original and the rescaled predictions.

The total proteome level of a pathway is usually dominated by a few highly abundant proteins. To ascertain that the growth rate-dependent trends of pathways were not caused by a few highly abundant proteins, we normalized measured protein abundance measurements to a *z*-score: for all proteome fractions for a given protein, we subtracted the mean proteome fraction for the protein across conditions and divided by the corresponding standard deviation. To obtain a normalized estimate of proteome allocation changes across conditions, we defined the normalized proteome allocation in a given condition as the sum of these *z*-scores for all pathway proteins. We found that the normalized proteome allocation is correlated significantly with the total proteome allocation across conditions (Pearson's $r^2 \geq 0.51$, $P \leq 0.004$ except for "others"; Fig. S6). Thus, the growth rate dependence of proteome allocation to a pathway is a joint phenomenon of the pathway proteins and is not dominated by a few highly abundant proteins. While the *in vivo* enzyme turnover number estimates are highly stable across

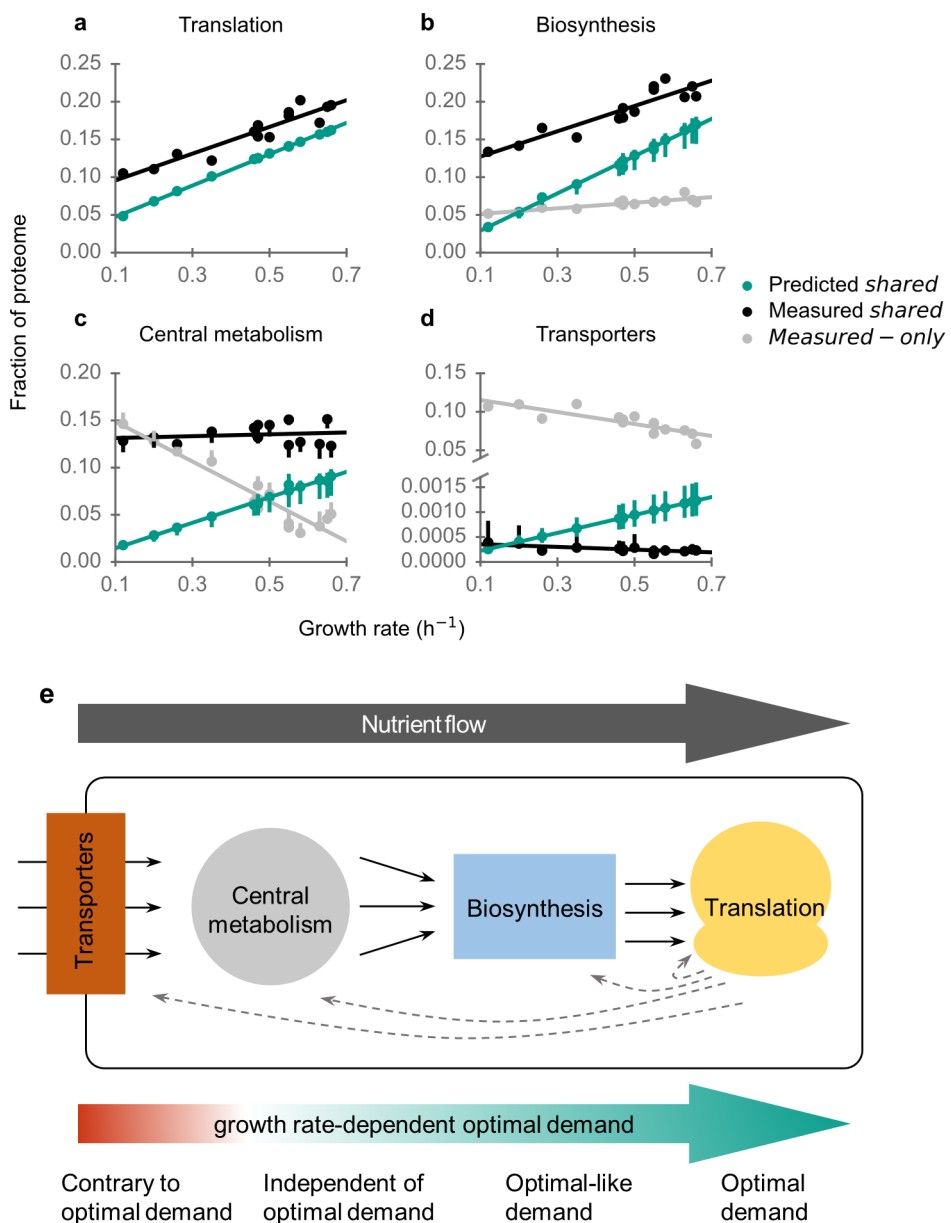

**FIG 1** Growth rate-dependent proteome efficiency increases along the nutrient flow. Predicted and observed proteome allocation to (a) translation machinery, (b) biosynthesis pathways, (c) central metabolism, and (d) transporters. Error bars in (b)–(d) extend from the 5th percentile to 95th percentile of 100 simulations, each with randomly perturbed turnover numbers. (e) Schematic diagram of nutrient flow and proteome efficiency.

different types of experiments (30, 31), the *in vitro* $k_{cat}$ estimates are rather noisy (9). To test the robustness of our predictions, we performed 100 simulations where we added random, normally distributed noise to the reciprocal of individual turnover numbers $k_i$ (see Materials and Methods). We found that the predictions are stable to variations in the turnover number estimates (error bars in Fig. 1 to 3), except for energy generation (Fig. 3).

Proteome allocation to translation is near the optimal prediction (Fig. 1a and Table 1), with no production of unneeded proteins ($f_{measured-only} = 0$), a very high correlation between observed and predicted total investment across conditions ($r_{pathway}^2 = 0.87$), a mean deviation between predicted and observed individual protein concentrations of only 35% (GMFE$_{pathway}$ = 1.35), and a strong correlation between observed and

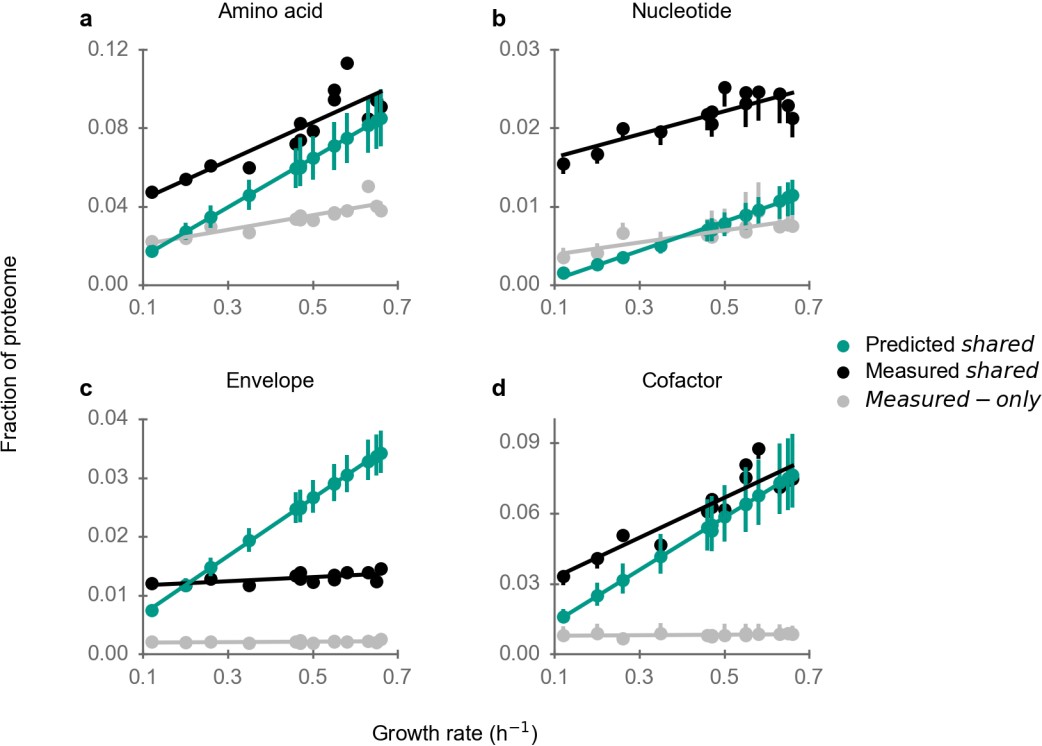

**FIG 2** Experimentally observed and predicted proteome fractions of biosynthesis pathways across glycolytic carbon sources. See Fig. S5b for biosynthetic proteins not covered here. Error bars extend from the 5th percentile to 95th percentile of 100 simulations, each with randomly perturbed turnover numbers.

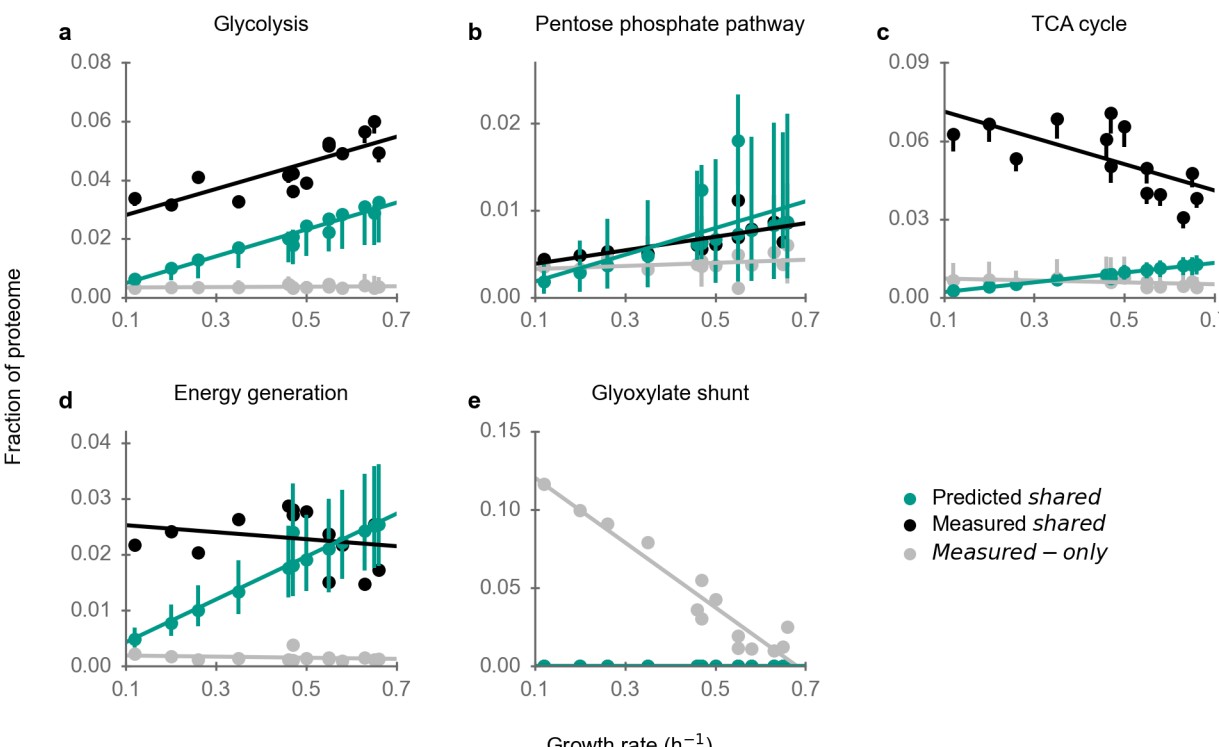

**FIG 3** Experimentally observed and predicted proteome fractions of central metabolic pathways. Error bars extend from the 5th percentile to 95th percentile of 100 simulations, each with randomly perturbed turnover numbers. See Fig. S5c for central metabolic proteins not covered here.

predicted individual protein concentrations (median across the 14 glycolytic conditions: $r_{individual}^2 = 0.98$). The remaining discrepancy between measured and predicted data is largely caused by the presence of deactivated ribosomes and elongation factor Tu at the studied growth rates (20), which cannot be predicted by optimization.

Proteome allocation to biosynthesis pathways is quantitatively consistent with the predictions for shared proteins, i.e., those whose presence is both predicted and observed (Fig. 1b; $r_{pathway}^2 = 0.84$; $GMFE_{pathway} = 1.70$; $r_{individual}^2 = 0.45$). However, about a quarter of the biosynthesis protein mass present in the cell is not predicted ($f_{measured-only} = 0.26$).

In central metabolism, the abundance of shared proteins is almost constant across growth rates in measured data, whereas it should increase with growth rate according to the predictions (Fig. 1c). Remarkably, the abundance of measured-only proteins is very high at low growth rates and decreases sharply with growth rate.

In stark contrast to all other pathways, the vast majority of transporters—more than 90%—are measured-only, i.e., the experimentally observed proteins are not part of the predicted optimal proteome ($f_{measured-only} = 0.92$; Fig. 1d; see Materials and Methods for the treatment of carbon transporters). Moreover, proteome allocation to transporters decreases with growth rate in measured data (both shared and measured-only), whereas it increases with growth rate in the locally optimal predictions ($r_{pathway} = -0.75$, $P = 1.9 \times 10^{-3}$). We note that when the concentration of a substrate is the limiting factor for cell growth, the optimal proteome allocation to its transporter increases with decreasing growth rate (10). Here, to compare transporters across growth conditions differing by the available carbon source, we excluded all proteins annotated as transporters for the carbon nutrients used in any of the studied conditions. Since these carbon sources are the only nutrients that vary in abundance across growth conditions (17), the data considered here are for the non-growth-limiting transporters, and their abundance indeed scales contrary to optimal demand. The true deviation from optimality may be smaller than this estimate due to the existence of many alternative transporters (25) and due to inaccurate turnover number estimates for transporters; only 24 out of 774 transport reactions have experimentally measured turnover numbers.

A large mass fraction of the proteins that cannot be assigned to one of the pathways described above (others) is also not expected to be present in the cell according to our predictions ($f_{measured-only} = 0.91$; Fig. S7a). About 40% of this unexpected protein mass is related to degradation pathways. At the same time, the abundance of shared proteins is similar to the predictions ($GMFE_{pathway} = 1.79$).

In sum, proteome efficiency increases along the nutrient flow in the four coarse-grained pathways (Fig. 1e). Transporters represent the metabolic interface of the cell to the environment. In the absence of external sensors, the presence of a transporter for a potential nutrient is a necessary condition for its detection by the cell; thus, non-optimal transporter abundance serves an important cellular function unrelated to steady-state growth. Central metabolism acts as a hub that connects all other pathways. When nutrients are transported into the cell, they either directly enter central metabolism, or they first need to be degraded by catabolism. For this reason, optimal proteome allocation to central metabolism is strongly environment-dependent. Just as is the case for transporters, keeping a certain fraction of central metabolism enzymes in standby for environmental changes will thus be beneficial in transitions between physiological states. Moreover, the optimal abundance of central metabolism proteins would require detailed, environment-dependent regulation, which may be difficult to achieve without substantial cellular investment into sensing and regulation. In contrast, optimal resource allocation into translation and the biosynthesis (anabolic) pathways, which synthesize building blocks for the cell, is largely independent of nutrients across minimal environments and depends almost exclusively on the growth rate. Their optimal regulation is thus a one-dimensional problem that requires only a sensor for growth rate itself and can be implemented relatively easily. Consistent with this speculation, biosynthesis and translational genes are regulated by fewer transcription factors than transporters and

central metabolic genes (Fig. S8). At the same time, our observations are consistent with a reserve of unused biosynthesis enzymes at low growth rates (Fig. 1a and b), which can benefit the cell in fluctuating conditions (32, 33).

## The most expensive biosynthesis pathways are consistent with optimality

To find if proteome efficiency varies in biosynthesis, we further divided biosynthesis pathways into five sets of pathways: amino acid biosynthesis; nucleotide biosynthesis; cofactor biosynthesis; cell envelope component biosynthesis; and all other biosynthesis enzymes. The predicted proteome fractions of these pathways are almost linear functions of the growth rate (Fig. 2), as mostly the same reactions are expected to be used for biosynthesis across the studied minimal conditions.

A large fraction of the proteome is allocated to amino acid biosynthesis pathways at high growth rates on minimal media (about 15%, Fig. 2a). Similar to the situation for translation, proteome allocation to amino acid biosynthesis pathways is strongly correlated with predictions (Fig. 2a; $r_{pathway}^2 = 0.77$; $GMFE_{pathway} = 1.40$; $r_{individual}^2 = 0.45$; Table 1). However, in contrast to translation, a sizeable proteome fraction for amino acid biosynthesis is invested into proteins not predicted to be active ($f_{measured-only} = 0.30$).

For nucleotide biosynthesis pathways, predicted and observed abundances of shared proteins are also strongly correlated in ($r_{pathway}^2 = 0.67$), but their magnitudes differ by more than threefold ($GMFE_{pathway} = 3.32$; Fig. 2b). Moreover, the abundance of individual enzymes in this pathway cannot be explained well by the predictions ($r_{individual}^2 = 0.15$).

Cell envelope biosynthesis pathways encompass lipid, peptidoglycan, and LPS biosynthesis. While predicted and observed abundances of shared enzymes in these pathways show a statistically significant correlation ($r_{pathway}^2 = 0.43$; Table 1), the slopes of their growth rate dependences differ markedly. The observed proteome allocation is almost constant across growth conditions; in contrast, the predicted proteome allocation increases proportionally with growth rate (Fig. 2c). It is noteworthy that this disagreement does not stem from an incorrect assumption of constant biomass composition across conditions: our model explicitly accounts for the changing biomass fractions of cell envelope components (see Materials and Methods), which are in particular due to changes in cell size. That predictions substantially exceed observed proteome allocation for cell envelope biosynthesis at faster growth may reflect cellular non-optimality, but would also be consistent with an erroneous assignment of low turnover numbers to one or more enzymes. Additionally, we note that the predictions of biosynthesis pathways for amino acids, nucleotides, and cell envelope components are largely unaffected by the reformulated growth rate-dependent biomass composition, as all results obtained under the assumption of a constant biomass composition are consistent with those observed in the original calculations (Fig. S9).

Similar to amino acid biosynthesis pathways, cofactor biosynthesis pathways are also highly abundant at high growth rates (about 10% of the total proteome, Fig. 2d). Proteome allocation to cofactor biosynthesis pathways is highly consistent with the optimal predictions ($r_{pathway}^2 = 0.84$; $GMFE_{pathway} = 1.24$; $f_{measured-only} = 0.11$; $r_{individual}^2 = 0.59$). Furthermore, when considering the saturation of enzymes (Fig. S5), the experimentally measured abundance of amino acid and cofactor biosynthesis pathways falls between the original predicted shared proteome (a lower bound for the optimal proteome) and the rescaled predictions with enzyme saturation (an upper bound for the optimal proteome). This further supports the notion that the proteome allocation to amino acid and cofactor biosynthesis pathways is regulated for maximal efficiency.

In sum, proteome efficiency varies substantially across biosynthesis pathways. While observed proteome investment only increases by roughly twofold for amino acid, nucleotide, and cofactor biosynthesis and shows almost no increase in envelope and other biosynthesis pathways, predicted investment increases by almost a factor of 5.5 (which is the fold-change of growth rate across the examined conditions). At lower

growth rates, we expect decreasing enzyme saturation (29) and thus a progressively stronger underestimation of the required proteome by the model; accordingly, Fig. 2a and d appear to be highly consistent with an optimal abundance of the shared proteins of amino acid and cofactor biosynthesis pathways. On the other hand, proteome allocation to nucleotide, envelope, and other biosynthesis pathways (Fig. S7b) appears to be sub-optimal.

## Central metabolism: precursor metabolite and energy generation pathways appear not to be regulated for optimality

The enzymes of central metabolism show little systematic variation with growth rate, and their abundance is at most weakly correlated with the predicted concentrations ($r_{\text{pathway}}^2$ = 0.024; $\text{GMFE}_{\text{pathway}}$ = 2.32). To examine if individual pathways show a stronger agreement between observations and predictions, we examined six central metabolic pathways: glycolysis; pentose phosphate pathway; tricarboxylic acid cycle (TCA cycle); energy generation pathways, comprising the electron transport chain and ATP synthase; glyoxylate shunt; and other central metabolic enzymes.

Proteome allocation to glycolysis increases markedly with growth rate and is strongly correlated with predicted values (Fig. 3a; $r_{\text{pathway}}^2$ = 0.63; $f_{\text{measured-only}}$ = 0.08). Protein levels are substantially higher than predicted ($\text{GMFE}_{\text{pathway}}$ = 2.21), although considering enzyme saturation leads to a convergence between predicted and observed proteome fractions at the highest growth rates (Fig. S5). A potential reason for this discrepancy is that most of the reactions in glycolysis are reversible, while the simple approximation for enzyme activity used here ($k_{\text{cat}}$) cannot capture the demand of enzymes close to thermodynamic equilibrium (34). Moreover, many of the enzymes in glycolysis are regulated allosterically (35), and may hence act at lower activities than assumed in the simulations.

The pentose phosphate pathway also shows significant signs of partial optimality: the measured abundance of shared proteins is close to and strongly correlated with the predictions (Fig. 3b; $r_{\text{pathway}}^2$ = 0.72; $\text{GMFE}_{\text{pathway}}$ = 1.3). However, measured-only proteins account for 39% of the pathway proteome.

Enzyme abundance in the TCA cycle is decidedly non-optimal. The abundance of shared enzymes decreases with growth rate, while predictions indicate it should increase (Fig. 3c; $r_{\text{pathway}}$ = −0.65). In addition, enzyme abundance is massively higher than predicted across all growth rates ($\text{GMFE}_{\text{pathway}}$ = 6.4). At the same time, measured-only proteins account for only a very small fraction of the pathway ($f_{\text{measured-only}}$ = 0.10), and the abundances of individual proteins are also correlated with measured data ($r_{\text{individual}}^2$ = 0.38, $P$ = 0.03).

The proteome fraction allocated to energy generation pathways—comprising the electron transport chain and ATP synthase—is almost independent of the growth rate, while predictions increase with growth rate (Fig. 3d). Similar to the TCA cycle, measured-only proteins make up only a small fraction of the pathway (6%). *E. coli* fully oxidizes carbon sources to $CO_2$ at low growth rates under aerobic conditions (aerobic respiration), while at high growth rates it only partially oxidizes some carbon sources—in particular glucose and fructose—resulting in the excretion of acetate (aerobic fermentation, leading to overflow metabolism). Along with the metabolic switch from aerobic respiration to aerobic fermentation, the TCA cycle is gradually downregulated (36). In our predictions, aerobic fermentation is more efficient than aerobic respiration for all conditions, so that only aerobic fermentation was active in the predictions. However, even with a model that predicted the switch to fermentation, our conclusions would likely not change; this is because the switch would not affect lower growth rates and because the predicted demand into the TCA cycle would only change slightly.

We were surprised to find that the proteins of the glyoxylate shunt (comprising AceA, AceB, and GlcB) are highly abundant at low growth rates (~12% of the proteome at $\mu$ = 0.12 $h^{-1}$; Fig. 3e), with a proteome fraction almost twice that of its alternative pathway, the TCA cycle (Fig. 3c). This high abundance at low growth rates does not appear to

be specific to the BW25113 strain, as it is mirrored in the MG1655 strain (Fig. S10a) (3, 37). Fluxomics data show that across many conditions with low growth rates, flux into the glyoxylate shunt is roughly equal to the flux into the TCA cycle (38–43) (Fig. S10b). In contrast, the model predicts the glyoxylate shunt to be inactive except in growth on acetate.

In sum, proteome allocation to the pathways of central metabolism is not well explained by optimal proteome efficiency alone, at least not as far as can be discerned with the type of model employed here. This is particularly true for the metabolic switches from aerobic respiration to aerobic fermentation and from the glyoxylate shunt to the TCA cycle.

## Utilization of alternative pathways cannot be explained by optimal proteome efficiency

With increasing growth rate, metabolic fluxes may shift between alternative pathways. For example, energy production from glucose switches from aerobic respiration to aerobic fermentation (overflow metabolism) (36). Consistent with previous studies (38, 40), we found that with increasing growth rate, flux gradually transitioned from the phosphoenolpyruvate (PEP)-glyoxylate cycle to the TCA cycle (Fig. S10).

Neither aerobic respiration nor the glyoxylate shunt is used in the predicted flux distributions. In constraint-based models, overflow metabolism emerges when a previously redundant, additional growth-limiting constraint becomes active (44). While there is evidence that overflow metabolism is rooted in a limit on proteome investment into catabolic enzymes (36, 45), this effect cannot be reproduced in mechanistic models without corresponding empirical adjustments. For example, one way of enforcing aerobic fermentation is to impose a decrease in proteome usage and an increase in energy production with increasing growth rate (36, 46); another is to allocate a constant empirical mass of proteins to energy production (47).

The PEP-glyoxylate cycle, which contains the glyoxylate shunt, represents an alternative route to the TCA cycle (38). Compared to the TCA cycle, the PEP-glyoxylate cycle produces an additional NADH instead of one NADPH (38). Since NADPH is a common cofactor in anabolic pathways in *E. coli*, it was suggested that the cell should choose the pathway which can produce more NADPH (the TCA cycle) at high growth rates (38). However, the interconversion between NADPH and NADH is a very common process in *E. coli* (48), and it is not clear how the small difference in pathway output (1 NADPH vs 1 NADH) could explain the massive resource allocation (~12% of the proteome) into the glyoxylate shunt at low growth rates. Recent studies showed that overexpression of the genes encoding glyoxylate shunt enzymes can reduce the lag time when *E. coli* experiences a transition from a glycolytic carbon source to a gluconeogenic carbon source (49, 50). However, it is still challenging to develop mechanistic models that explain the growth rate-dependent proteome allocation to alternative pathways and lag times from first principles.

## Proteome efficiency can explain the proteome allocation to metabolic pathways in evolved strains

The growth rate of evolved strains tends to increase during long-term lab evolution experiments (19). This phenomenon is likely caused to a large extent by an adaptive shift to a more optimal proteome allocation pattern. We thus expect that the pathways with near-optimal abundance in unevolved strains will be upregulated in evolved strains to sustain higher growth rates, while overabundant pathways will be downregulated. To test our hypothesis, we used a recent paper which measured the transcriptome of eight *E. coli* strains adapted to continuous exponential growth on a minimal glucose medium (51).

Since wild-type *E. coli* already has a relatively high growth rate on glucose, the abundances of most pathways are close to optimal in wild-type *E. coli* (Fig. 2 and 3). As expected, in evolved strains, more genes are upregulated than downregulated in

most pathways; these include all biosynthesis pathways as well as the glycolysis and energy generation pathways (Fig. S11). In contrast, proteins in the TCA cycle and "others" (proteins without an assigned pathway in this study) are overabundant in wild-type *E. coli*. As expected, in evolved strains, more genes are downregulated than are upregulated in both of these pathways. While transporters are also predicted to be overabundant in the wild type, more transporter genes are upregulated than downregulated in evolved strains (21.8% vs 19.0%; two-sided paired *t*-test, $P = 0.27$), the opposite of what was expected. It may be that this result is a consequence of lumping all transport proteins into one "pathway", regardless of which types of molecules are transported. We speculate that the predicted trends might become more evident in *E. coli* adaptation to less-preferred carbon sources. For example, while *E. coli* has a much lower growth rate on galactose than on glucose (0.26 h$^{-1}$ vs 0.58 h$^{-1}$), the pathway usage is the same for these two carbon sources, except for the additional enzymes required for galactose degradation.

## Conclusions

In this study, we systematically assessed proteome efficiency at the pathway level across *E. coli* growing on minimal media with different carbon sources. Overall, we found that the proteome efficiency of pathways increases along the nutrient flow, from transporters to central metabolism to biosynthesis pathways to translation. We note that this gradient is analogous to a gradient of genomic stability observed on much longer time scales, with central reactions being more stable over evolutionary time than reactions at the interface to the environment (52), which we found here to also be less efficient. Above, we showed that proteome allocation is near the optimal demand for the most expensive biosynthesis pathways, including translation as well as amino acid and cofactors biosynthesis pathways; the same pathways are located in the interior of the cellular biosynthetic network. In contrast, about half of the metabolic pathways by mass show a growth rate dependence contrary to that expected for optimal demand, including the TCA cycle, glyoxylate shunt, and transporters; typically, these pathways are located at the periphery of the cellular network. We hypothesize that these patterns of local optimality and sub-optimality arise from two tradeoffs and their interactions: on the one hand, the tradeoff between maximal instantaneous growth and the cell's ability to quickly and efficiently transition its physiological state in response to environmental changes, and on the other hand, the tradeoff between the benefits of precise and optimal control of cellular resource allocation and the resource investment required for the corresponding control systems. Quantifying these tradeoffs and their joint influence on cellular physiology will require an enhanced, quantitative understanding of the evolutionarily relevant patterns of environmental changes as well as of the costs and effectiveness of regulatory strategies available to bacteria such as *E. coli*.

## MATERIALS AND METHODS

### Growth rate-dependent biomass composition

The original biomass composition in the *i*ML1515 model is very similar to that of the *i*AF1260 model, formulated for a doubling time of 40 min or $\mu = 1.04$ h$^{-1}$ (53). However, biomass composition varies across growth rates. The two most significant changes are those of the RNA/protein mass ratio and the cell volume, which determines the surface/volume ratio (*S/V*). Both ratios can be expressed as functions of the growth rate; accordingly, we estimated the growth rate-dependent biomass fraction of RNA, protein, and cell envelope components (including murein, lipopolysaccharides, and lipid) as functions of the growth rate, as described below.

We first fitted experimental data for the RNA/protein mass ratio ($\frac{m_{\mathrm{RNA}}}{m_{\mathrm{protein}}}$) (4, 54) and the surface/volume ratio (*S/V*) (28) to linear functions of the growth rate (Fig. S1), resulting in the relationships as follows:

$$\frac{m_{\text{RNA}}}{m_{\text{protein}}}(\mu) = 0.223\mu + 0.08 \,, \tag{1}$$

$$\frac{S}{V}(\mu) = -0.1895\mu + 7.952 \,. \tag{2}$$

Assuming that the biomass contribution of cell envelope components ($m_{\text{envelope}}$) is proportional to the surface/volume ratio gives

$$\frac{m_{\text{envelope}}(\mu = \mu_1)}{m_{\text{envelope}}(\mu = \mu_2)} = \frac{\frac{S}{V}(\mu = \mu_1)}{\frac{S}{V}(\mu = \mu_2)} \,. \tag{3}$$

The growth rate-dependent biomass fraction of cell envelope components ($m_{\text{envelope}}$) can then be estimated by equation (3) given equation (2) and $m_{\text{envelope}}$ at $\mu = 1.04$ h$^{-1}$. The relative composition of murein, lipopolysaccharides, and lipid was assumed to be constant.

The biomass fractions of cellular components other than RNA, protein, and cell envelope components ($m_{\text{others}}$) were assumed to be independent of the growth rate. The sum of RNA and protein is given as follows:

$$m_{\text{RNA}} + m_{\text{protein}} = 1 - m_{\text{others}} - m_{\text{envelope}} \,. \tag{4}$$

Combining equations (1) and (4), the content of RNA and protein can be calculated for all conditions (Fig. S1). The relative contributions of individual nucleotides to total RNA and of individual amino acids to total protein were assumed to be growth rate-independent. The resulting growth rate-dependent biomass compositions are listed in Table S1.

## Implementation of MOMENT

We used ccFBA (26) for all simulations, which implements the MOMENT algorithm (9) with improved treatment of co-functional enzymes (27). We obtained maximal *in vivo* effective enzyme turnover number ($k_{\text{app,max}}$) values determined across evolved *E. coli* strains by Heckmann et al. (31). For enzymes and transporters for which $k_{\text{app,max}}$ was unavailable, we used *in vitro* $k_{\text{cat}}$ values collected by Adadi et al. from public databases (9). When both $k_{\text{app,max}}$ and $k_{\text{cat}}$ were unavailable, we used maximal *in vivo* enzyme turnover numbers predicted from machine learning methods ($k_{\text{app,ml}}$) by Heckmann et al. (31). Finally, for transporters that could not be parameterized by $k_{\text{app,max}}$, $k_{\text{cat}}$, or $k_{\text{app,ml}}$, we used an approximate value of 65 s$^{-1}$ as suggested by reference (27). The sources and values of turnover numbers are listed in Table S2. Reactions with missing turnover numbers were parameterized with the median of all other turnover numbers in the model.

To test the robustness of our model, we perturbed the turnover numbers through random sampling. The turnover numbers $k_i$ enter the model results through the enzyme concentrations, which are proportional to $1/k_i$. Sampling $1/k_i$ from a symmetric distribution leads to an expectation value for the enzyme concentration required to carry 1 unit of flux that is equal to that at the original $k_i$; this would not be the case if we sampled $k_i$ from a symmetric distribution around the original $k_i$. To avoid statistical biases, it is thus preferable to perturb to $1/k_i$ rather than $k_i$. For each reaction $i$ with turnover number $k_i$, we assumed that the perturbed $1/k_i$ follows a normal distribution with a mean of $1/k_i$ and a standard deviation $\sigma = 0.3\frac{1}{k_i}$. In each of 100 independent iterations, for each reaction $i$, we drew a random ($k_i^{\text{rand}}$) value. We avoided extreme values by restricting variation to at most a factor of 100, i.e., when a randomly generated $k_i$ was >100 times larger or <0.01 times smaller than the original $k_i$, we replaced the random value with the original $k_i$. In each iteration, we then estimated the locally optimal proteome using the perturbed turnover numbers. The interval from the 5th percentile to

the 95th percentile of the estimates across the 100 iterations are shown as error bars in Fig. 1 to 3; Fig. S4 and S7.

The standard application of constraint-based methods such as MOMENT is to maximize the growth rate in a given nutrient condition. Because we are instead interested in the locally optimal proteome allocation at the observed growth rate, we solved the complementary optimization problem that estimates the minimal required proteome ($C$) able to support the observed growth rate on the given carbon source. However, as the objective function in ccFBA is the growth rate, we used an indirect procedure for the solution. Due to the linear problem formulation of ccFBA, there is a linear relationship between proteome investment and the predicted growth rate, $C = a\mu + b$ with two constants $a$ and $b$. Note that due to a non-zero non-growth-related maintenance energy term included in the model, $b > 0$. The constants $a$ and $b$ can be determined from any two pairs of proteome budget $C$ and growth rate $\mu$.

For each experimental condition with observed growth rate $\mu'$ according to reference (17), we first estimated the biomass composition at $\mu'$. At this biomass composition, we then predicted the growth rates at $C = 0.1$ g/g$_{DW}$ ($C_{0.1}$) and $C = 0.2$ g/g$_{DW}$ ($C_{0.2}$), denoted as $\mu_{0.1}$ and $\mu_{0.2}$, respectively. $a$ and $b$ were then calculated from $\mu_{0.1}$, $\mu_{0.2}$, $C_{0.1}$, and $C_{0.2}$. The total minimal required proteome ($C'$) at the observed growth rate was then read out as $C' = a\mu' + b$.

For a given protein $i$, its optimal demand at the observed growth rate $\mu'$ ($p_{i,\mu}$) in units of g/g$_{DW}$ can be expressed as follows:

$$p_{i,\mu'} = \frac{C'}{C_{0.1}} p_{i,\mu_{0.1}}. \tag{5}$$

with the $p_{i,\mu_{0.1}}$ minimal demand for protein $i$ at $C_{0.1}$.

With the protein content in dry mass at $\mu'$ ($m_{\text{protein},\mu'}$) estimated in equation (4), the proteome fraction of protein $i$ at $\mu'$ ($m_{i,\mu'}$) can be written as follows:

$$m_{i,\mu'} = \frac{p_{i,\mu'}}{m_{\text{protein},\mu'}}. \tag{6}$$

## Estimation of enzyme saturation level

As both the *in vivo* concentrations of metabolites and the corresponding Michaelis constants $K_m$ are not available on a large scale, we estimated an expected saturation level for each enzyme in a given growth condition as a function of the growth rate. In previous work, we found that at a given reaction flux, the concentrations of an enzyme and its substrate in *E. coli* are such that their combined mass density is minimal (29). For a reaction following irreversible Michaelis-Menten kinetics, the resulting optimal substrate concentration ($[S]^*$) can be written as (29)

$$[S]^* = \sqrt{\frac{aK_m\upsilon^*}{k_{\text{cat}}}}, \tag{7}$$

where $\upsilon^*$ is the reaction flux, $k_{\text{cat}}$ is the enzyme turnover number, and $a$ is the molecular weight ratio between the enzyme and the substrate. Under the reasonable assumption that most reaction fluxes scale proportionally with the growth rate, $\upsilon \propto \mu$, this equation can be simplified to

$$[S]^* = [S]_{\text{ref}}\sqrt{\frac{\mu}{\mu_{\text{ref}}}}, \tag{8}$$

where $[S]_{\text{ref}}$ is the optimal substrate concentration in a reference state with growth rate $\mu_{\text{ref}}$ [to see how equation (8) follows from equation (7), replace $\upsilon = c\mu$ with some

constant $c$, and then divide equation (7) by a version of itself that is parameterized for the reference state].

We chose growth on a minimal glucose medium as the reference state, with $\mu_{glc} = 0.58$ h$^{-1}$. In this condition, the metabolite concentration is typically twice its corresponding $K_m$ (29). For each reaction, we hence set the substrate concentration in the reference state to $[S]_{glc} = 2K_m$.

With $\mu_{glc} = 0.58$ h$^{-1}$, $[S]_{glc} = 2K_m$, and the measured $\mu$, the expected substrate concentration can be calculated by equation (8). From this, we obtained the expected enzyme saturation level ($f_{sat}$) as follows:

$$f_{sat} = \frac{[S]^*}{[S]^* + K_m} = \frac{2K_m\sqrt{\mu/\mu_{glc}}}{2K_m\sqrt{\mu/\mu_{glc}} + K_m} = \left(1 + \frac{1}{2}\sqrt{\frac{\mu_{glc}}{\mu}}\right)^{-1}; \qquad (9)$$

note that the enzyme-specific value for $K_m$ cancels out, so that the expected saturation level becomes the same for all enzymes. Numerically, the saturation level according to equation (9) increases from 0.45 at $\mu = 0.1$ h$^{-1}$ to 0.69 at $\mu = 0.7$ h$^{-1}$.

We re-did all simulations using rescaled effective turnover numbers $k_{i,rescaled} = k_{cat} \times f_{sat}$, with $k_{cat}$ the original turnover number. The rescaled predictions are shown in Fig. S5.

## Pathway membership

Proteins were characterized as transporters if the corresponding genes were assigned to transport processes according to the *i*ML1515 annotation (25). The carbon source is the only nutrient that differs between the minimal media used in the proteomics experiments (17). To make the transporters comparable across conditions, we thus excluded inner and outer membrane transporters for all carbon sources used in the studied conditions (17) and analyzed only the transporters for other metabolites.

We used the pathway ontology in EcoCyc (55) (downloaded on 13 January 2021) to assign the enzyme members for other metabolic pathways.

Proteins are labeled as biosynthetic enzymes based on the EcoCyc pathway ontology annotation "biosynthesis" (55). The pathways included in this category are as follows: (1) amino acid biosynthesis ("Amino Acid Biosynthesis" in EcoCyc), (2) nucleotide biosynthesis ("Nucleoside and Nucleotide Biosynthesis"), (3) cofactors ("Cofactor, Carrier, and Vitamin Biosynthesis"), and (4) cell envelope components ("Cell Structure Biosynthesis and Fatty Acid and Lipid Biosynthesis"), including lipid, peptidoglycan, and LPS. All other biosynthetic enzymes are merged into (5) other biosynthetic pathways. See Table S5 for the corresponding hierarchy levels in the EcoCyc pathway ontology.

Enzymes are designated as being involved in precursors and energy generation according to the EcoCyc pathway ontology annotation "Generation of Precursor Metabolites and Energy". Pathways in this category are as follows: (1) glycolysis (2), pentose phosphate pathways (3), TCA cycle (4), glyoxylate bypass (EcoCyc does not list a pathway for the glyoxylate shunt; the three genes classified as glyoxylate shunt are aceA, aceB, and glcB) (5), energy production ("Electron Transfer Chains and ATP Biosynthesis"), and (6) other enzymes.

## *Treatment of enzymes involved in the nucleotide salvage pathway*

In the range of studied growth rates, the transcription of mRNA accounts for more than half of the total RNA transcription (1). The half-life of mRNA is very short (~5.5 min) (56) compared to the doubling time, and degraded mRNA will be reused through the nucleotide salvage pathway. However, our model only predicts the proteome allocation to *de novo* biosynthesis pathways. To make the prediction comparable with the observed data, the nucleotide salvage pathway was thus excluded from "nucleotide biosynthesis pathway".

## Transcriptional regulation data

Experimental data sets of RegulonDB v10.9 (57) were used for counting the number of transcription factors regulating each protein.

## ACKNOWLEDGMENTS

We thank Hugo Dourado, Deniz Sezer, and Peter Schubert for helpful discussions.

This work was supported by the Volkswagen Foundation under the "Life?" initiative, and by the German Research Foundation (DFG) through grant CRC 1310. The funders had no role in study design, data collection and analysis, decision to publish, or preparation of the manuscript.

X.-P.H. conceived and designed the study and performed analysis. S.S. performed the analysis of transcriptional factor regulation. M.J.L. supervised the study. X.-P.H. and M.J.L. interpreted the results and wrote the manuscript.

The authors declare that no competing interests exist.

## AUTHOR AFFILIATIONS

[1]Institute for Computer Science, Heinrich Heine University, Düsseldorf, Germany
[2]Department of Biology, Heinrich Heine University, Düsseldorf, Germany

## AUTHOR ORCIDs

Xiao-Pan Hu http://orcid.org/0000-0003-4461-4613
Martin J. Lercher http://orcid.org/0000-0003-3940-1621

## FUNDING

| Funder | Grant(s) | Author(s) |
| --- | --- | --- |
| Volkswagen Foundation (VolkswagenStiftung) | "Life?" initiative | Martin J. Lercher |
| Deutsche Forschungsgemeinschaft (DFG) | CRC 1310 | Martin J. Lercher |

## AUTHOR CONTRIBUTIONS

Xiao-Pan Hu, Conceptualization, Data curation, Formal analysis, Investigation, Methodology, Validation, Visualization, Writing – original draft, Writing – review and editing | Stefan Schroeder, Formal analysis | Martin J. Lercher, Funding acquisition, Supervision, Writing – review and editing

## ADDITIONAL FILES

The following material is available online.

### Supplemental Material

**Supplemental material (mSystems00760-23-S0001.pdf).** Fig. S1-S11 and Table S1-S6.

### Open Peer Review

**PEER REVIEW HISTORY (review-history.pdf).** An accounting of the reviewer comments and feedback.

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
