## [Reviewer comments · mSystems]

Proteome efficiency of metabolic pathways in *Escherichia coli* increases along the nutrient flow

Xiao-Pan Hu, Stefan Schroeder, and Martin Lercher

Corresponding Author(s): Martin Lercher, Heinrich-Heine-Universität Dusseldorf

Review Timeline:

Submission Date:

July 25, 2023

Accepted:

August 24, 2023

Editor: William Harcombe

Reviewer(s): The reviewers have opted to remain anonymous.

Transaction Report:

DOI: <https://doi.org/10.1128/mSystems.00760-23>

August 24, 2023

Prof. Martin J. Lercher
Heinrich-Heine-Universität Dusseldorf
Institute for Computer Science
Düsseldorf
Germany

Re: mSystems00760-23 (Proteome efficiency of metabolic pathways in *Escherichia coli* increases along the nutrient flow)

Dear Prof. Martin J. Lercher:

Your manuscript has been accepted, and I am forwarding it to the ASM Journals Department for publication. For your reference, ASM Journals' address is given below. Before it can be scheduled for publication, your manuscript will be checked by the mSystems production staff to make sure that all elements meet the technical requirements for publication. They will contact you if anything needs to be revised before copyediting and production can begin. Otherwise, you will be notified when your proofs are ready to be viewed.

If you would like to submit a potential Featured Image, please email a file and a short legend to msystems@asmusa.org. Please note that we can only consider images that (i) the authors created or own and (ii) have not been previously published. By submitting, you agree that the image can be used under the same terms as the published article. File requirements: square dimensions (4" x 4"), 300 dpi resolution, RGB colorspace, TIF file format.

We recognize that the video files can become quite large, and so to avoid quality loss ASM suggests sending the video file via <https://www.wetransfer.com/>. When you have a final version of the video and the still ready to share, please send it to mSystems staff at msystems@asmusa.org.

Sincerely,

William Harcombe
Editor, mSystems

Journals Department
E-mail: mSystems@asmusa.org